# IL-1 Superfamily Across 400+ Species: Therapeutic Targets and Disease Implications

**DOI:** 10.3390/biology14050561

**Published:** 2025-05-17

**Authors:** Weibin Wang, Dawei Li, Kaiyong Luo, Baozheng Chen, Tingting Hao, Xuzhen Li, Dazhong Guo, Yang Dong, Ya Ning

**Affiliations:** 1College of Science, Yunnan Agricultural University, Kunming 650201, China; wangwb@dongyang-lab.org (W.W.);; 2Yunnan Provincial Key Laboratory of Biological Big Data, Yunnan Agricultural University, Kunming 650201, China; dli@dongyang-lab.org (D.L.);

**Keywords:** IL-1 superfamily, evolution, therapeutic targets, disease implications, cross-species analysis

## Abstract

The IL-1 superfamily represents crucial immune regulators implicated in various diseases. Through comprehensive evolutionary analysis across 400+ animal species, we identified highly conserved functional motifs (“F-F” and “FES-PG-WF”) that exhibit remarkable structural stability throughout evolution. These preserved regions, maintained over millions of years, likely represent critical functional domains. Our phylogenetic reconstruction reveals lineage-specific expansion patterns that parallel immune system sophistication. The conserved structural elements provide ideal targets for rational drug design, offering the potential for developing precise therapeutics against inflammatory diseases, autoimmune disorders, and malignancies. These findings establish an evolutionary foundation for structure-based immunotherapy development against this clinically significant cytokine family.

## 1. Introduction

Cytokines belonging to the IL-1 superfamily are proteins with a molecular weight of approximately 17–18 kDa. They feature a characteristic β-trefoil pyramidal barrel structure (β-trefoil fold) formed by six two-stranded β-hairpins [1]. This structural motif is shared by numerous proteins across evolution, from bacteria to mammals, and is often associated with diverse functions, particularly carbohydrate recognition (such as ricin-type lectins) and toxins [2,3,4]. In the human body, the IL-1 system is mainly composed of the following functional proteins. Agonist activity: IL-1α, IL-1β, IL-18, IL-33, IL-36α, IL-36β, IL-36γ; receptor antagonists: IL-1Ra, IL-36Ra, IL-38; anti-inflammatory cytokine: IL-37; and receptors acting as signaling molecules: IL-1R1, ST2, IL-18Rα, IL-36R; accessory proteins: IL-1RAcP, IL-18Rβ; decoy or negative regulatory receptors: IL-1R2, SIGIRR. Finally, the system includes receptors that are still considered orphan or whose function is poorly defined: TIGIRR-2, TIGIRR-1; and IL-18BP [5,6] (Figure 1). IL-1 family proteins/subunits are represented in this article by IL-1s, whereas IL-1 family receptor proteins/subunits are represented by IL-1Rs.

Protein structure similarity has long been regarded as compelling evidence of common ancestry. However, the beta-trefoil fold, a hallmark of the IL-1 ligand superfamily, is widespread across many unrelated proteins in diverse species. This suggests that various amino acid sequences can independently adopt this structure through distinct evolutionary events rather than solely through shared ancestry [4,9]. The genes, protein sequences, and structures of the IL-1 family have been conserved throughout evolution, suggesting that they play crucial roles in an organism’s ability to adapt and survive [10]. According to previous research, teleost fish have homologs of IL-18 and its probable receptor complex, indicating that some IL-1 superfamily members first appeared early in the evolution of vertebrates [11]. Researchers have verified the origin of these genes by analyzing the genomes of several metazoans. They also suggested that all IL-1s and IL-1Rs most likely originated from a single ancient ancestor rather than being directly linked to the advent of mammals [10,12]. Furthermore, a comparison of the genomes of humans, mice, and pufferfish showed that only a small number of mammalian IL-1 family members have distinct fish homologs, indicating that certain mammalian IL-1 family members may have developed more recently [11].

Human IL-1 family genes are predominantly located in a gene cluster on chromosome 2 (2q14.1), facilitating the coordinated regulation of their expression [1,13]. Based on gene localization and sequence similarity across multiple species, three distinct IL-1 subfamilies were identified: IL-1, IL-18, and IL-33. Although IL-18 and IL-33 are not evolutionarily related to the IL-1 subfamily, they share structural and functional similarities [12]. Phylogenetic analysis revealed that IL-1Ra, IL-1α, IL-1β, IL-36α, IL-36β, IL-36γ, IL-36Ra, IL-37, and IL-38 are family members with a common ancestry. The ancestral proto-IL-1β gene emerged approximately 420 million years ago (Mya), coinciding with the emergence of the vertebrate subphylum [12]. The IL-1α and IL-33 subfamilies are exclusive to mammals, appearing after the divergence of the Sauropsida (ancestral lineage of reptiles, turtles, and birds) and Synapsid (ancestral lineage of mammals) clades around 320 Mya and before the divergence of mammalian species around 160 Mya [12,14,15]. In contrast, the IL-18 subfamily, like IL-1β, is expressed in all vertebrates and likely originated around 420 Mya [14]. The IL-1α cluster likely emerged from a duplication event around the appearance of the Synapsid proto-mammalian clade. Despite the sequence-level divergence between IL-18, IL-33, and IL-1α from IL-1β, chromosomal gene anatomy and intron/exon structural homology strongly suggest that IL-1α arose from a gene duplication of IL-1β between 320 and 160 Mya, while IL-18 and IL-33 originated from separate evolutionary events [10,16,17]. There is strong evidence from sequence analysis and chromosomal anatomy that IL-1R1, IL-1R2, IL-1RAcP, ST2, IL-18Rα, IL-36R, and IL-18Rβ are members of the same family and originate from ancestral gene duplications of the common proto-IL-1R. Specifically, IL-1RAcP likely arose from a duplication event in IL-18Rβ [12]. Given that all members of the IL-1R superfamily, except TIGIRR-2, are present in all vertebrates, it is probable that these genes diverged before the separation of bony and cartilaginous fish (~420 Mya). TIGIRR-1 likely formed from a duplication and translocation event involving a member of the IL-1R1 subcluster. In contrast, TIGIRR-2 probably originated from gene duplication of TIGIRR-1′s ancestor before the divergence of bony fish and the Tetrapoda clade (~365 Mya) [14,18].

Significant interspecies differences in the distribution, gene copy number, function-related sequence features, and three-dimensional structural aspects of IL-1s and IL-1Rs have been observed across a wide range of evolutionary positions [11,12]. These comparative genomic studies have elucidated the evolutionary conservation, structural divergence, and functional diversification of IL-1s and IL-1Rs across distinct phylogenetic lineages, providing crucial insights into their molecular evolution and functional plasticity [10,19]. Of particular significance is the growing body of research exploring the pathophysiological implications of IL-1 family genes in various disease states, including autoimmune disorders, chronic inflammatory conditions, cardiovascular pathologies, and oncogenesis [20,21,22]. Emerging research frontiers include the intricate interplay between IL-1 family cytokines and the host microbiome [23], their immunomodulatory roles within the tumor microenvironment (TME) [24,25,26], their potential as immunotherapeutic targets [26], and their involvement in metabolic disorders such as obesity and metabolic syndrome [27]. Despite the identification of IL-1 family homologs across multiple taxa, the functional characterization and evolutionary trajectories of these cytokines in non-mammalian vertebrates remain poorly elucidated [10,11]. Furthermore, the precise evolutionary chronology and molecular mechanisms underlying the functional diversification of IL-1 family members require further investigation [10,19]. Consequently, a more comprehensive understanding of IL-1 family biology, particularly through comparative phylogenetic analyses, could facilitate the development of targeted therapeutic strategies and pharmacological interventions directed at these cytokines or their cognate receptors [20,21,28]. This could help address IL-1s/IL-1Rs-related disease. These include fulminant myocarditis (FM) [29], chronic obstructive pulmonary disease (COPD) [30,31], Alzheimer’s disease (AD) [32], multiple sclerosis (MS), rheumatoid arthritis (RA), systemic lupus erythematosus (SLE), psoriasis, Sjogren’s syndrome, inflammatory bowel disease (IBD) [33], coronary artery disease [34], chronic heart failure [35,36], depressive symptoms [37], and head and neck squamous cell carcinoma (HNSCC) [38]. In particular, it may facilitate the selection of animal models for drug testing, creation of new medications, drug delivery systems, drug screening and optimization, multi-target drug design, and immunotherapy [39].

Here, we used phylogenetic analysis to examine the distribution, evolutionary patterns, and sequence features of IL-1s and IL-1Rs in more than 400 species. This approach provides new insights into the origins, species distribution, key mutations, and potential anti-inflammatory drug targets in animals.

## 2. Materials and Methods

### 2.1. Construction of the Species Tree

Databases such as NCBI, Ensembl, CNCB, and Macgenome were the main sources of the genomes of 518 individual animals (Appendix A), including GFF annotation files. These datasets were released before 30 June 2023, and contained complete annotation information for 246, 265, 6, and 1 individuals, respectively. mammalia_odb10 was used as the database, and the buscophylo (https://github.com/ypchan/buscophylo, accessed on 14 May 2025) script was used to evaluate the genome assembly quality and extract informative statistics from the Benchmarking Universal Single-Copy Orthologs (BUSCO) [40] ortholog (protein) sequences. Species with C (complete) > 2.4%, S (single-copy) ≥ 0.60%, D (fragmented) ≥ 0.10%, F (fragmented) ≥ 0.50%, and M (missing) ≤ 97.00% were retained, and those for which the CDSs could not be extracted from the GFF files were excluded. Single-copy orthologous sequences were aligned using MAFFT v7.525 [41] with specific parameters (--genafpair--maxiterate 1000) to optimize alignment accuracy. The aligned sequences were trimmed using trimAl v1.5 [42] to remove poorly aligned positions and highly divergent regions (-gt 0.85 -cons 30). Trimmed alignments were used to construct a phylogenetic tree with IQtree v2.3.6 [43], and maximum-likelihood phylogenies were constructed using the JTT  +  F  +  R10 substitution model with 1000 bootstrap replicates. Multiple phylogenetic trees were merged using Astral v5.7.8 [44] to generate a species tree. *Sphagnum magellanicum* was retained as the outgroup for animal phylogenetic analysis.

### 2.2. Preparation of Genome and Annotation Files

The formats of the genome and GFF files were universally standardized. The AGAT Toolkit [45] was used to extract the longest isoform of the candidate species for genome and annotation file pre-processing. The GFF3 files that satisfied the requirements were sorted by processing the GFF files, where the script agat_convert_sp_gxf2gxf.pl was used to find and correct errors and missing information in the GFF files. The longest isoform was preserved via the agat_sp_keep_longest_isoform.pl. All the CDS and protein sequences of the candidate species were extracted in batches via GffRead v0.12.7 [46]. Finally, the CDS and protein sequences of the candidate species were combined into a single file.

### 2.3. Identification and Phylogenetic Analysis of the IL-1s and IL-1Rs

To identify all IL-1s and IL-1Rs, we used the query sequences (proteins) of human IL-1s and IL-1Rs and their key domains (Appendix A). A database was built by pfam_scan-1.6 [47] with default parameters, based on the combined protein file as indicated above, and members of distant species protein families were discovered using HMMER-3.3.2 [48]. To identify protein sets that contained several key domains, the intersection of the search results for each key domain was considered. To obtain more precise protein family information, we additionally employed the BLASTp [49] identification method with an E-value threshold of 1 × 10^−5^ to identify homologous sequences. The intersection of Pfam and BLASTp identification results was used to determine the final identified protein set. Finally, the IL-1s and IL-1Rs family trees were created using the same strategy as the species tree described above. To obtain conservative sequence information for the candidate sequence, a custom-written R-script was used to extract domain and gene structure information from the Pfam search results and gene location in the GFF file.

### 2.4. Protein Family Clustering and Classification

All data visualization work was performed using R 4.4.1, and dgfr v0.0.0.9 [50] was used to determine the ideal number of k-means clustering groups for protein sequences (min_clust = 4, max_clust = 10) and calculate the mean/median similarity within each cluster. Ggtree v3.12 [51] and ggtreeExtra v1.14 [52] were used to enhance the aesthetics of phylogenetic trees. Gggenes v0.5.1 [53] was used for visualizing motifs, domains, and gene structures. MSA v1.36.1 [54], with the multiple sequence alignment (MSA) algorithm MUSCLE v5.1 [55], was used for visualizing sequence alignment. Protein subfamilies with evident structural traits were grouped after they were found and thoroughly clustered using k-means clustering, motif, domain, and gene structure characteristics.

### 2.5. Analysis of the IL-1s and IL-1Rs Variants

Weblogo v3.7.12 [56] was used to perform a SeqLogo analysis of the sequences of various groups. ESMfold (https://github.com/facebookresearch/esm, accessed on 14 May 2025), a high-precision protein structure prediction tool, is utilized for the structural prediction of regrouped protein sequences based on the pre-trained model esm2_t48_15B_UR50D [57]. US-align was used for the universal structure alignments of predicted structures across all redefined groups [58]. Finally, UCSF ChimeraX v1.9 [59] was employed for the visualization and aesthetic appeal of the structure alignment results.

## 3. Results

### 3.1. Identification and Distribution of IL-1s and IL-1Rs in Animal Class

We constructed an evolutionary tree spanning approximately 1290 million years, encompassing 491 individual animals in a phylogenetic tree, including 480 animal species and an *S. magellanicum* outgroup. The species ultimately selected ranged from higher to lower taxa, including Mammalia, Aves, Actinopteri, Lepidosauria, Reptilia, and other Class. After obtaining the intersection of gene/protein sets identified by Pfam seed and BLASTp identification methods, we acquired an evolutionary tree of 2490 IL-1s from 402 individuals (396 species) (Figure 2A,B) and an evolutionary tree of 2700 IL-1Rs from 413 individuals (407 species) (Figure 2C,D).

We classified all IL-1s and manually grouped them into seven groups, Groups I to VII (Figure 2B), by considering identity, distinct separation features PCA (Figure 2E) results, and phylogenetic relationships. Among these, the homologs of the IL-36 subfamily (IL-36α, IL-36β, and IL-36γ) (Group VII) and IL-1α (Group III) are exclusively distributed in mammals, showing a distribution pattern similar to that previously reported [12], and they exhibit high identity compared to humans (Figure 2A). IL-36-like proteins were entirely located in cluster 4 and partially in cluster 2 (Figure 2A,B), with their homologs traceable as far back as *Ornithorhynchus anatinus* in Monotremata, showing identity ranging from 34.65% to 42.14% compared to humans. Notably, cluster 3 (Group VI) includes almost all IL-38 and IL-1Ra homologs (Figure 2B), with homologs distributed in species as distant as *Sphenodon punctatus* in Lepidosauria (38.10–49.66%). The homologs of IL-1β (Group IV, V) are traceable to *Scleropages formosus* and *Paramormyrops kingsleyae* in Actinopteri (27.54–30.62%), with Group V proteins not found in mammals and Actinopteri, representing a distinct IL-1β subfamily (terrestrial oviparous vertebrates) widely distributed in Aves (Figure 2F). Additionally, IL-18, IL-33, and IL-1α are almost entirely located in cluster 1 but are not similar to each other, forming separate clusters classified as Group I, II, and III. Group II (IL-33) and Group III (IL-1α), like Group VI and Group VII, are exclusively distributed in mammals (Figure 2F,G). The homologs of IL-1α, IL-33, and IL-18 were traceable to *Naja naja* in Lepidosauria (25.66%), *Tachyglossus aculeatus* in Monotremata (27.78%), and *P. kingsleyae* in Actinopteri (29.95%). For IL-1Rs, we employed the same strategy, dividing them into seven groups, Group I to VII (Figure 2D). The clustering analysis of different IL-1R sequences based on PCA shows a clear separation between clusters, indicating that different clusters have significantly distinct features in the principal component space (Figure 2H). The homologs of IL-1Rs share varying degrees of similarity. Among these, the homologs of IL-18Rα, IL-18β, IL-1R2, and ST2 are traceable to *Callorhinchus milii* in Chondrichthyes (27.83–44.91%), while IL-1RAcP and SIGIRR can be traced back to *Erpetoichthys calabaricus* in Cladistia (47.96%) and *Eptatretus burgeri* in Myxini (39.70%), respectively. The homologs of IL-1R1 and IL-36R are traceable to *Gekko japonicus* in Lepidosauria (39.86–43.71%) (Figure 2I).

### 3.2. Evolutionary of IL-1s and IL-1s Across Several Groups

The IL-18 and IL-1β homologs in teleosts are considered the ancestral form of the IL-1 superfamily in animals (Figure 2B,F). However, in the immune system of teleosts, IL-18 homologs are virtually absent, while IL-1β homologs (except for Group V) are highly abundant (Figure 2F,G). This suggests that *IL18* likely evolved after animals fully transitioned from aquatic to terrestrial environments. IL-1β homologs are a major component of the IL-1 immune system in teleost. Broadly, as mentioned above, IL-1β homologs can be divided into two subgroups: non-mammalian IL-1β (Group V, terrestrial oviparous vertebrates) and mammalian IL-1β (Group IV). These two groups of proteins exhibited significant differences in both evolutionary relationships and species distribution (Figure 2B,F). As shown in Figure 2G, and in conjunction with the preliminary conclusions by Rivers-Auty et al. [12] regarding the origin of IL-1 superfamily proteins, we further divided the origin of the IL-1 superfamily into three stages, from ancient times to the present: ancient IL-1s (IL-1β, IL-18), paleo-IL-1s (IL-1Ra, IL-38), and neo-IL-1s (IL-1α, IL-37, IL-36, IL-33). Among them, the IL-1β subfamily may have undergone more than two divergences, which we speculate occurred around the same time as the divergences in Group VI.

IL-1Rs are unique, seemingly existing as low-copy genes since the emergence of vertebrates. As marine life began to spread out on land during the Amphibia epoch, their gene copy numbers abruptly increased significantly. This is not the same for IL-1s. The Reptilia epoch showed an increase in the number of IL-1 gene copies, whereas Aves had very few IL-1 gene copies (Figure 2G and Figure 3A,B). This suggests that IL-1 ligands gradually adapted to IL-1 receptors throughout the evolutionary process. Despite this, it is evident that, apart from mammals, SIGIRR (Group I) and IL-1RAcP (Group V) homologous genes are also widely distributed in Actinopteri, indicating that these are relatively ancient IL-1Rs genes (Figure 2I). ST2 (Group II) appears after amphibians, likely representing a more recently evolved IL-1Rs gene (Figure 2I). Another notable point is that IL-1R2 (Group III) seems to be unique to mammals, with very few or no homologs in other species (Figure 2I). For instance, no IL-1R2 homologs have been found in Aves, and the specific reasons are unclear. Similar to IL-1RAcP, IL-1R1 homologs (Group IV) are abundant in mammals, making them the most prevalent IL-1 receptor gene in mammals. IL-18Rβ (Group VI) and IL-18Rα (Group VII) exhibited the closest evolutionary relationship, with highly similar distribution patterns (Figure 2D,I). They were tandemly located on the same chromosome, suggesting that *IL18RAP* and *IL18R1* originated from a gene duplication event. *IL1R1*, *IL1R2*, *IL18R1*, *IL1RL2*, *IL1RL2*, and *IL18RAP* formed a cluster on the same chromosome, indicating gene duplication events, as suggested previously [12]. Despite this, the sequence identity between IL-1RAcP and both IL-18Rβ and IL-18Rα was modestly low (Figure 2B). *SIGIRR* does not cluster on a single chromosome position; previous studies have shown that *SIGIRR* has an independent evolutionary history [12].

Phylogenetic evidence indicates that IL-1α, IL-33, and IL-36 subfamily homologs represent mammalian-specific innovations originating ~160 million years ago (Mya), with IL-1α and IL-36 genes arising from IL-1β duplication events. In contrast, IL-38 emerged concurrently with terrestrial oviparous vertebrate IL-1β variants, potentially as an adaptive response to Triassic mass-extinction environmental pressure. Strikingly, IL-1Rs exhibit far deeper evolutionary roots—their origin likely predates previous estimates of 420 Mya, extending beyond the vertebrate subphylum. This was substantiated by our identification of high-identity SIGIRR homologs (39.70% identity) in Myxini (hagfish), suggesting advanced receptor maturation, even in these basal vertebrates. Other IL-1R subfamilies presumably underwent subsequent diversification, possibly during early vertebrate genome duplication.

### 3.3. Main Cross-Species Characteristics of IL-1s and IL-1Rs

In IL-1s, groups I, VI, and VII consist of a single IL1 domain, whereas groups III, IV, and V primarily contain an IL1 domain along with an IL1_propep (an ~115-amino acid N-terminal propeptide cleaved to release active IL-1s), although a few members retain only the IL1 domain (Figure 4). In contrast, Group II was mainly composed of IL33_bt. Among groups I, II, VI, and VII, species with shorter IL1 or IL33_bt domains (<70 aa) were predominantly small mammals such as *Myotis lucifugus*, *Procavia capensis*, and *Sorex araneus* (Figure 4). Meanwhile, in Group III, only three species—*Choloepus hoffmanni*, *Manis javanica*, and *Monodelphis domestica*—lacked the IL1_propep domain, retaining only a single IL1 domain (Figure 4). A similar pattern was observed in groups IV and V (both IL-β homologs), where most members were found in Actinopteri and Aves, with only a few mammals (e.g., *Erinaceus europaeus*, *Mustela putorius furo*, and *Notamacropus eugenii*) exhibiting this structure (Figure 2F). Notably, within Actinopteri, IL-β homologs containing the IL1_propep domain are restricted to *Cyclopterus lumpus* and *Clupea harengus*. In contrast, mammals exhibited the highest abundance of IL-β homologs in this domain, far surpassing those in Aves and other primitive species (Appendix A). This suggests that the IL1_propep domain may have been gradually acquired during evolution and subsequently expanded in higher mammals. Additionally, this domain is closely associated with Caspase-1 mediated pathways, despite its absence in Group I. In summary, all IL-1 ligand family members—except IL-33—contain at least one IL1 domain, while IL-33 is uniquely composed of the IL33_bt domain. This distinction implies that the *IL33* gene may have originated independently of other IL-1s genes, emerging later in evolution and around the time of mammalian divergence.

The IL-1Rs demonstrate complex domain organization across its subgroups, with the extracellular C-terminal portions typically containing 0–3 immunoglobulin domains (ig, Ig_2, Ig_3, or Ig_6) and occasionally featuring specialized I-set domains (Ig-like) crucial for ligand recognition and binding (Figure 4). Group I receptors with most SIGIRR-like proteins containing only a single Ig-like domain or, in many cases, solely the intracellular TIR domain without any extracellular components–a pattern widely distributed across Class though notably rare in Aves, with only *Calypte anna* showing detectable Ig_3 domains (Figure 4). Within this group, Ig domains show taxon-specific distributions. While Ig_3 appears broadly in mammalian SIGIRRs, the ig domain is uniquely conserved in *O. anatinus*, and Ig_2 predominates in Actinopteri and mammalian lineages, with *Oryzias latipes* and *Oryzias melastigma* exhibiting the distinctive C1-set domain variant (Appendix A). Group II receptors reveal phylogenetic constraints, with I-set domains exclusively present in mammals (particularly primates) and Aves (Figure 2I), mirroring their roles in cell adhesion molecules such as neural cell adhesion molecules (CAM) [60], although many members in these Class retain only TIR domains. The Group III members show striking absence in avian species while being widely distributed across mammals and appearing sporadically in basal vertebrates like Lepidosauria and Chondrichthyes (Figure 2I), typically lacking TIR domains, except in putative ancestral forms found in *C. milii* and *Lepisosteus oculatus*. All other subgroups conserve the TIR domain (Figure 4)—a ∼50 amino acid motif exhibiting extraordinary evolutionary conservation from plants to mammals that is critical for signaling and shows key functional residues like P446 determining ligand specificity [61,62]. Group IV receptors demonstrate particular structural diversity, typically containing two Ig-like domains (Figure 4) with reptilian variants uniquely possessing I-set domains (Appendix A), while phylogenetic evidence suggests shared ancestry between Groups III/IV subfamilies (*IL1R1*, *IL1R2*, *IL1RL1*, and *IL1RL2*) despite Group IV’s remarkable diversification (Figure 2D). The unexpected 28.14–29.85% sequence identity between Group V and Group I (SIGIRR), along with Group V’s 34 SIGIRR-like genes predominantly found in Actinopteri and Mammalia (all cluster 4 members, see Figure 2C: marked by star), challenges previous evolutionary models by suggesting that ancestral *SIGIRR* may have given rise to *IL18R1*, *IL18RAP*, and *IL1RAP*. This extensive structural and evolutionary variation, particularly in Ig domain composition and arrangement, underscores why IL-1-targeted therapies developed for humans may show limited efficacy across other vertebrate species, owing to fundamental differences in receptor architecture.

Although cytokines typically contain a clear signal sequence [63], IL-1 family ligands lack hydrophobic N-terminal signal peptides [64]. These absent N-terminal sequences may influence intracellular protein localization and trafficking [65], forcing most IL-1 members to utilize non-classical secretion pathways, although the precise mechanisms remain unclear [66]. Our signal peptide predictions revealed that IL-1Ra (Group VI) possesses a unique N-terminal hydrophobic signal peptide for efficient secretion (Appendix A). Unlike IL-1β, pre-IL-1Ra undergoes spontaneous cleavage and secretion after the removal of its ~25-residue hydrophobic signal peptide, with the signal peptide-containing isoform becoming active extracellularly [64]. However, exceptions exist: Our results show that only 23.43% of IL-1Ra isoforms contain signal peptides, predominantly in mammals (Appendix A), while signal peptide-negative isoforms resemble other IL-1 ligands retained intracellularly [64]. For instance, the IL-36Ra precursor includes a 23-residue signal peptide. SeqLogo analysis showed that Group III exhibits the most conserved propeptide sequences, followed by Groups II and IV (Appendix A), aligning with their evolutionary conservation (Figure 2B). Overall, merely 6.50% of IL-1s across species harbor N-terminal signal peptides, nearly exclusively in terrestrial chordates with sophisticated immune systems, suggesting that these signal peptides represent evolutionarily acquired features likely enabling specialized functions.

Most IL-1 extracellular domains adopt the β-trefoil fold, which is a highly conserved structural feature [67]. Through spatial structure prediction and alignment, we found that nearly all IL-1 family ligand proteins across diverse species retain this conserved β-trefoil architecture, typically featuring a conserved internal hydrophobic cavity (Figure 3C)—a characteristic previously confirmed in both chicken and human IL-1β [68]. Only a few species exhibited irregular or incomplete β-trefoil fold variants (Figure 3C). Although β-trefoil-fold proteins may originate from diverse gene types [2] and display low sequence conservation [69], their structures are largely similar. To date, only a few conserved amino acid residues have been reported within the IL-1s β-trefoil fold [70], and they are all based on IL-1 β. However, we identified structurally conserved hydrophobic residues, notably the “F-F” and “FES-PG-WF” motifs (Figure 3C,D and Appendix A). These adjacent motifs appear to be universally conserved across all seven subgroups and may play critical roles in maintaining structural and functional stability, as they remain virtually unchanged throughout prolonged, cross-species evolution.

### 3.4. Key Targets of IL-1s

Key residue and motifs are one of the critical factors that influence the function of drug targets. Through MSA and SeqLogo analysis, we identified multiple cross-species conserved signature motifs, key residues, cleavage sites, and species-specific non-conserved residues across 402 animal species (Appendix A).

All Group I proteins possess a highly conserved, cross-species caspase-4-cleavage site characterized by the N-terminal “LESD” motif (Appendix A: Group I). Similar to the K89 that affects the function of IL-18, this residue remains almost unchanged in all vertebrates. In Group II, the nuclear localization signal (NLS) domain (including the chromatin-binding motif), the thrombin cleavage site R48, and the caspase 3/7 cleavage sites D178, all of them exhibit significant cross-species conservation (Appendix A: Group II). The cleavage sites for inflammatory proteases (residues 65–112), and thrombin cleavage site R106 [71] exhibit limited conservation and display distinct species-specific characteristics (Appendix A: Group II). In Group III, the NLS, histone acetyltransferase-binding (HAT-binding) domains of pro-IL-1α are highly conserved across mammalian species. Our findings demonstrate that these domains are conserved in nearly all mammalian species (Figure 2F). Moreover, there is no evidence to suggest that these two types of conserved motifs evolved in species before mammals. Instead, they appear to be unique to mammals, this conclusion contradicts previous speculations [72].

This also includes the well-characterized key residues of IL-1β—including D116 [73,74], R4, R11, Q15, H30/33, L31/34, Q32, F45, K103/104 [68,70,75,76,77], F157, E113, Y121 and so on (Appendix A: Group IV/V)—exhibit strong cross-species conservation and are closely linked to structural and functional integrity. Although our research identified a distinct type of IL-1β homolog that shares homology with human IL-1β but diverges significantly from mammalian and teleost IL-1β orthologs (Figure 2B,F), current functional studies of these residues through site-directed mutagenesis have been largely restricted to teleosts and mammals (Group IV). Terrestrial oviparous vertebrates (Group V) remain poorly characterized (Figure 2F and Appendix A: Group V), and our findings suggest that they represent a more primitive IL-1β isoform.

Beyond the known functional residues, IL-1s universally share two defining structural features: (1) a conserved β-trefoil fold containing the signature “F-F” and “FES-PG-WF” motifs (Figure 3C,D and Appendix A), with E113 being the only currently identified functionally critical residue within these motifs [75], and a hyperconserved C-terminal F157 flanked by semi-conserved D145, Y147, and E150 (Appendix A: Group IV). These signatures reveal an evolutionary blueprint; while receptor-binding specificity is primarily mediated by N-terminal residues, the β-trefoil core maintains structural integrity and modulates binding activity.

## 4. Discussion

### 4.1. The Ancient Origin of IL-1s and IL-1Rs

The current understanding of IL-1 family origins primarily derives from vertebrate studies, particularly the fish-to-mammal transition. This study conducts an extensive analysis of the IL-1s and a limited analysis of the IL-1Rs based on the currently sequenced animal species, thereby filling the gap in this area. Beyond their canonical roles in innate immunity and inflammation, IL-1s likely contribute to the development of adaptive immunity in vertebrates [1]. Phylogenetic analysis revealed progressive structural complexity: early vertebrates possess simplified IL-1 variants with IL-1α-like transcriptional regulation and processing, whereas reptilian orthologs exhibit IL-1β-like biological properties [78]. This functional divergence may reflect environmental adaptations in aquatic and terrestrial ecosystems. Given that the variations of IL-1s are always associated with the evolutionary events of vertebrates (Figure 2F,I), Its origin should coincide with the emergence of early vertebrates, approximately ~420 Mya [12]. Notably, lineage-specific gene duplications have expanded immune regulatory capacity—exemplified by the three tandem *IL1RL1* (*IL1RL1α*, *IL1RL1β*, *IL1RL1γ*) paralogs in rainbow trout (resembling mammalian IL-36 subfamily expansion in this study) [19] and the nine-gene IL-1 cluster on human chromosome 2 originating from IL-1β duplication events [12]. Our analyses indicate that IL-1s achieved structural and functional maturity early in chordate evolution, with significant family expansion and diversification occurring during the amphibian-to-mammalian transition (Figure 3A,B). This study categorizes the process into three distinct phases: the ancient IL-1s phase (IL-1β and IL-18), the paleo-IL-1s phase (IL-1Ra and IL-38), and the neo-IL-1s phase (IL-1α, IL-37, IL-36, and IL-33). Notably, the IL-1β subfamily is believed to have undergone more than two divergence events, which we hypothesize coincided with the divergence events in IL-38. Among these findings, the most noteworthy is terrestrial oviparous vertebrates (Group V), whose IL-1β subtypes exhibit a uniquely ultra-conserved evolutionary trajectory (Figure 2B). Intriguingly, this pattern is completely absent in mammals (Figure 2F), suggesting a distinct evolutionary pathway in this vertebrate group.

The IL-1Rs likely originated through duplication and diversification of a primordial IL-1R1-like ancestor [12], accounting for their shared structural features—particularly the conserved TIR domain (Figure 4). This evolutionary trajectory parallels that of Toll-like receptors (TLRs), which are quintessential innate immune receptors in invertebrates [5]. Both families exhibit striking functional and structural homology, with all members containing TIR domains [5]. Notably, immunoglobulin (Ig) domains—another hallmark of IL-1Rs—are also prevalent in invertebrate proteins [79,80,81], and IL-1R-like orthologs with vertebrate homology have been identified in Cnidaria, Protostomia, and Deuterostomia [80,81,82]. These findings suggest that IL-1Rs may have emerged during early vertebrate evolution, potentially coinciding with the 1R/2R whole-genome duplication events (~535.3–485.2 Mya). The presence of a SIGIRR homolog in *E. burgeri* with 39.70% identity suggests that this protein may have already possessed partial functional similarity to human SIGIRR at the time of this organism’s emergence. Consequently, its origin likely predates the previously estimated ~420 Mya, indicating a more ancient evolutionary history [12].

### 4.2. The Association of IL-1s and IL-1Rs with Diseases

IL-1s and IL-1Rs play critical roles in numerous human diseases, making them ideal therapeutic targets. Dysregulation of IL-1s pathways can lead to severe autoimmune and inflammatory diseases, including psoriasis, SLE, IBD [33,83], RA, type I diabetes, deficiency of the Interleukin-1 receptor antagonist (DIRA) [84,85], and so on. Current IL-1s- and IL-1Rs-related drugs targeting this pathway include Anakinra (recombinant IL-1Ra) [84,85], Canakinumab (anti-IL-1β monoclonal antibody), and Rilonacept (IL-1 trap fusion protein) [86]. However, their clinical use is limited by the risk of systemic immunosuppression, injection, and potential hepatorenal toxicity [86,87]. Current research on this family has primarily focused on mammalian IL-1β (Appendix A: Group IV), with limited studies on other ligands as potential therapeutic targets or their key cross-species conserved residues.

Cross-species drug research is crucial for precise and comprehensive drug design. Such as recent advancements in computational protein design have introduced novel mini-protein antagonists (minibinders), such as IL-1R1 and GP130 minibinders, which exhibit high affinity, high specificity, high thermal stability, short circulation time, and potential for local administration, making them highly effective and safe for treating cytokine storms while reducing the risk of long-term immune suppression; These agents, designed using Rosetta algorithms, effectively inhibited IL-1β-induced inflammation in human cardiac organoids, demonstrating the potential for localized delivery to mitigate systemic side effects [88]. The cross-species conserved and species-specific key residues/motifs identified in this study (Appendix A) may hold significant potential for high-throughput and AI-assisted drug development.

IL-1s and IL-1Rs are central to inflammatory and immune responses across vertebrates, yet most research has focused narrowly on mammalian IL-1β, leaving a critical gap in our understanding of their evolutionary conservation and functional diversity. Comparative genomic analyses across over 400 animal species have revealed striking patterns: while key residues in IL-1β (e.g., D116, H30/33, L31/34, F45, and K103/104) and IL-18 (e.g., E35/36, K53/89) are ultra-conserved across lineages (Appendix A), other functional domains—such as protease cleavage sites and receptor-binding interfaces—display species-specific adaptations. For example, the caspase-4-cleavage motif “LESD” in IL-18 and the thrombin-sensitive R48/R106 sites in IL-33 are evolutionarily conserved in mammals but diverge in other vertebrates (Appendix A), suggesting lineage-specific regulatory mechanisms. Similarly, IL-36 activation depends on proteolytic processing at species-variable N-terminal residues (K6/R5/S18), whereas IL-37 maturation depends on caspase-1 cleavage sites (D20, E21) that lack cross-species conservation (Appendix A). These findings highlight a fundamental dichotomy: a structurally conserved β-trefoil core (e.g., the “F-F” and “FES-PG-WF” motifs) ensures stability, while N-terminal and surface residues drive functional diversification (Figure 3C,D and Appendix A).

This evolutionary perspective has several implications. First, IL-1-targeting therapies effective in mammals do not translate to other species, particularly terrestrial oviparous vertebrates (Group V), which may retain primitive IL-1β isoforms with distinct receptor interactions (Figure 2F). IL-1β-targeting drugs effective in mammals and teleosts may lack efficacy in terrestrial oviparous vertebrates, except for a few shared conserved residues such as H30/33, L31/34, and the IL-1 family signature (Appendix A: Group V). Second, it identifies understudied lineages—such as birds and reptiles—as untapped models for probing ancestral immune mechanisms and developing agricultural interventions (e.g., poultry disease resistance). Third, the hyperconserved C-terminal motifs (e.g., F157, D145, and Y147) and nuclear localization signals (e.g., in IL-33 and IL-1α) suggest universal structural constraints that could be harnessed for broad-spectrum drug design (Appendix A). Conversely, lineage-specific adaptations offer precise therapeutic avenues. Critically, research on IL-1Rs lags far behind ligand studies, despite their equal importance. While sequence alignments revealed conserved extracellular domains (Appendix A), the structural complexity and length of IL-1Rs require functional validation—especially for non-mammalian species.

Bridging this gap could unravel how receptor-ligand co-evolution shapes immune responses across vertebrates, informing therapies for emerging zoonotic diseases, and conservation challenges. By integrating evolutionary biology with immunopharmacology, comparative studies of the IL-1 family can decode nature’s “blueprint” for balancing immune innovation with constraint, paving the way for both universal and lineage-tailored treatments.

### 4.3. Application Prospects and Limitations of Cross-Species Research

By identifying evolutionarily conserved motifs and species-specific variations, this research enables tailored therapeutics—from optimizing Anakinra-like drugs for veterinary use to designing cross-species vaccine adjuvants targeting IL-1′s receptor interface. It also provides information on zoonotic disease control and agricultural solutions. The conserved β-trefoil core offers a blueprint for broad-spectrum inhibitors, whereas lineage-specific domains guide precision medicine. Bridging evolutionary insights into biomedical needs, IL-1 cross-species analyses have unlocked strategies for immune modulation in health, agriculture, and conservation.

However, cross-species IL-1s and IL-1Rs research face the following key challenges: (i) Functional divergence—while sequence motifs are conserved, receptor binding and signaling often differ (e.g., primitive isoforms of Group V vertebrates may not respond to mammalian-targeting drugs). (ii) Knowledge gaps—most functional data are derived from mammals/teleosts, leaving most of the species experimentally unvalidated. (iii) Technical hurdles, such as the structural complexity of IL-1Rs and poor antibody cross-reactivity, limit mechanistic studies in non-model organisms. (iv) Translational risks: species-specific cleavage sites or immune contexts may render conserved targets ineffective in practice. Addressing this requires integrating CRISPR-based models, ancestral protein reconstruction, and multi-omics profiling across underrepresented clades.

In stark contrast to IL-1 ligands, research on IL-1Rs remains limited. While our work uncovered evolutionary conservation in their amino acid sequences (Appendix A), the extraordinary length and structural complexity of these receptors demand substantially more experimental evidence to validate their functional mechanisms.

## 5. Conclusions

This study provides a comprehensive evolutionary analysis of the IL-1 superfamily across over 400 animal species, revealing critical insights into the origins, conservation, and diversification of these key immune molecules. The identification of highly conserved motifs, such as the “F-F” and “FES-PG-WF” in the β-trefoil fold, underscores the structural stability and functional importance of these proteins throughout evolution. These conserved features, combined with lineage-specific variations, highlight the potential for both broad-spectrum and targeted therapeutic strategies. The ancient origins of IL-1 family members, dating back to early vertebrates, suggest that fundamental immune mechanisms have been preserved over millions of years. This knowledge not only enhances our understanding of the IL-1 superfamily’s role in health and disease but also provides a foundation for developing more effective and precise treatments for IL-1-related conditions. Future research should focus on further exploring the functional implications of these evolutionary findings and translating them into clinical applications, ultimately improving therapeutic outcomes and patient care.

## Figures and Tables

**Figure 1 biology-14-00561-f001:**
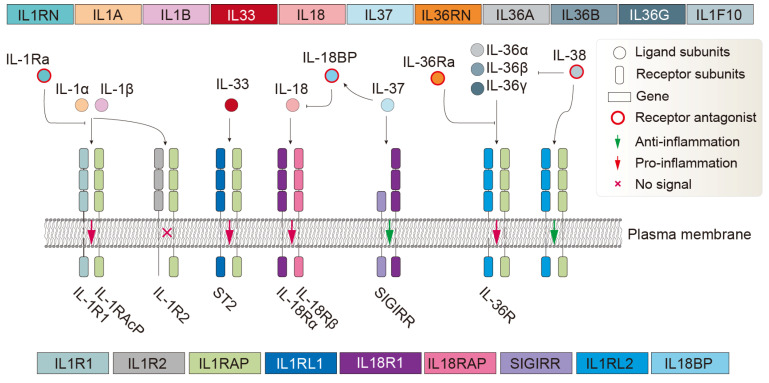
The ligand, antagonist, and receptor system of the human IL-1 family. The adjacent receptor subunits form dimeric complexes. The depicted receptor lengths do not reflect actual subunit lengths. While not all IL-1R family members have the typical structure of three extracellular Ig-like domains plus one intracellular TIR domain, this illustration uses the typical structure for representation. In reality, a receptor subunit may consist of 0–3 Ig-like domains (usually ig, Ig_2/3/6, C1/I-set), one transmembrane domain, and 0–1 TIR domain. IL-1R2, a ligand-binding chain for IL-1β and IL-1α, acts as a decoy receptor because it lacks the intracellular TIR domain and therefore can sequester ligands and IL-1RAcP into an inactive receptor complex [1]. The orphan receptors SIGIRR and TIGIRR-1/2 (not displayed) do not have recognized ligands [7], although it has been reported that IL-38 may bind TIGIRR-2 [8].

**Figure 2 biology-14-00561-f002:**
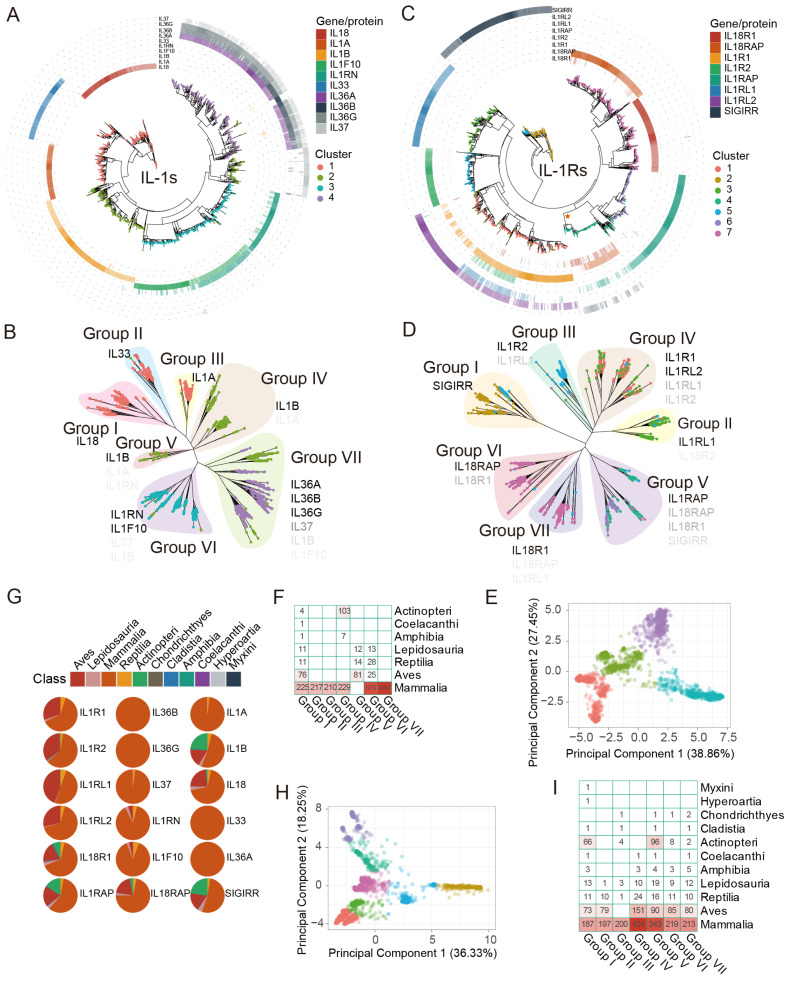
Phylogenetic relationship and distribution of IL-1s and IL-1Rs in more than 400 animal species. To ensure clarity and consistency, this figure utilizes HGNC (Hugo Gene Nomenclature Committee) gene symbol nomenclature instead of protein names for identification. This corresponds to Figure 1. The coloured dots in the PCA plot correspond to the tip points of the phylogenetic tree of IL-1s and IL-1Rs in terms of color. (**A**) Phylogenetic tree of IL-1s across 396 animal species, with different colored tip points representing different clusters and different colored rings outside the tree representing homologous proteins (Appendix A), with the depth of color corresponding to the identity (%) of query sequences of human IL-1s (Appendix A). Darker hues indicate greater identity. IL-1Ra was not included. (**B**) Phylogenetic tree and dispersion patterns of IL-1s. The depth of the symbol font color indicates the broad distribution of the main homologous gene/protein types in the groups, which are represented by gene symbols that match human homologous genes. The darker the color, the higher the proportion it represents in the corresponding group. (**C**) Phylogenetic tree of IL-1Rs across 407 animal species, similar to (**A**), IL-18BP, and TIGIRR-1/2 were not included. The clade marked with a star represents sequences that show similarity to the proteins encoded by the *SIGIRR*, *IL18R1*, *IL18RAP*, and *IL1RAP* genes. (**D**) Phylogenetic tree and dispersion patterns of IL-1Rs, similar to (**B**). (**E**) PCA shows the total number of clusters detected in IL-1s. These clusters exhibited significant differences in the original feature space. (**F**) The distribution of different groups of IL-1s at the Class level. (**G**) The percentage of IL-1s and IL-1Rs genes in Class. (**H**) PCA shows the total number of clusters detected in IL-1Rs, similar to (**E**). (**I**) The distribution of different groups of IL-1Rs at the Class level is similar to (**F**).

**Figure 3 biology-14-00561-f003:**
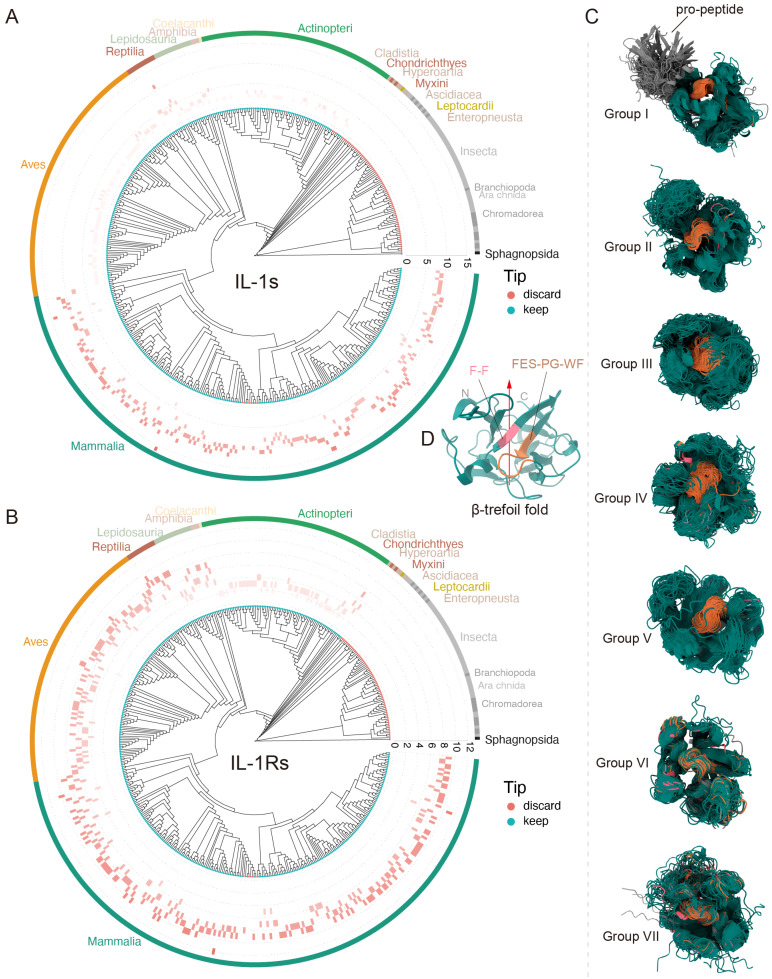
Conserved features during the evolution of IL-1s and IL-1Rs. (**A**,**B**) Gene copy number variation of IL-1s and IL-1Rs in more than 400 animal species. IL-1s underwent significant expansion during the mammalian period, whereas IL-1Rs maintained relatively stable copy numbers throughout all evolutionary stages. The tip points in cyan represent species individuals retained following BUSCO assessment, whereas the red points indicate the species that were filtered out; however, this does not mean that the filtered species were absent, but rather that the overall data did not meet the standards. The differently colored arcs in the outermost layer correspond to different Class. (**C**) Conserved 3D structures of IL-1s and IL-1Rs with non-conserved regions rendered 100% transparent. The β-trefoil fold and its ultraconserved motifs “F-F” and “FES-PG-WF” across species are represented by the colors dark teal, light pinkish red, and medium brown, respectively. The gray areas represent regions that are not conserved in sequence but are similar in structure (the propeptide region of Group I). (**D**), Localization of IL-1 family signatures within IL-1s monomeric protein (taking human IL-1α as a model). This structure corresponds to (C), where the red arrow indicates the opening of the β-trefoil fold, which is also the direction of the starting point of the C-terminus and the ending point of the N-terminus.

**Figure 4 biology-14-00561-f004:**
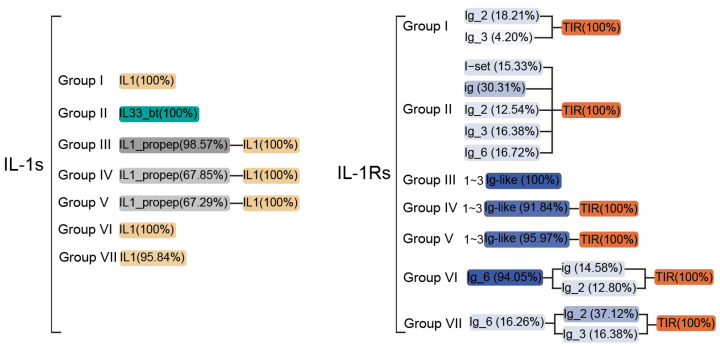
The composition of the main domains of IL-1s and IL-1Rs from over 400 different animal species in various groupings. Different colors represent different types of domains, while the intensity (transparency) of the color indicates the proportion of sequences containing the corresponding domain among the total sequences in each group; the darker the color, the higher the proportion. Domains with a proportion of less than 1% have been omitted from this figure. The Ig-like domain typically represents C1/I-set, ig, Ig_2/3/6. For more detailed information, please refer to Appendix A.

## Data Availability

The original contributions presented in the study are included in the Appendix A. Further inquiries can be directed to the corresponding authors.

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
