# Peer review of "IL-1 Superfamily Across 400+ Species: Therapeutic Targets and Disease Implications"

_biology, 2025, doi:10.3390/biology14050561_

Round 1
Reviewer 1 Report
Comments and Suggestions for Authors
In the manuscript by Wang et al titled ' IL-1 superfamily across 400+ species: therapeutic targets and disease implications', the authors have performed a detail analysis on the evolution of the IL-1 superfamily, identified lineage / species specific conservation and variation which provides an excellent template for development of target therapeutics for a plethora of IL-1 related disorders. I find this study sufficiently extensive and the conclusion made are well supported by the results. The speculation made by the authors based on their results throughout the manuscript make the manuscript more appealing and would be helpful for future research.
Although there have been similar studies on the IL-1 super-family in the past, I feel that this study digs deeper and hence holds significance. Two additional things that I can add are:
- Since the font size for some of the labels in Figure3 become extremely small and impossible to read a zoomed in inset of a region could be helpful.
- If the authors can demonstrate that a previously unknown drug-IL1 interaction can be established now due to the findings of this paper it would generate more interest..
Author Response
Comments 1: Since the font size for some of the labels in Figure3 become extremely small and impossible to read a zoomed in inset of a region could be helpful.
Response 1: Thank you very much for your positive and constructive comments on our manuscript. To improve clarity in Figure 3, we selectively removed unclear text (e.g., species names and non-essential Class labels) while enlarging critical elements, and updated Figure 3D to better represent IL-1 family ligand/receptor evolutionary patterns and key protein signatures (see Figure 3 and Lines 400-403).
Comments 2: If the authors can demonstrate that a previously unknown drug-IL1 interaction can be established now due to the findings of this paper it would generate more interest.
Response 2: Your suggestion regarding drug-IL-1 family interactions is highly appreciated and thought-provoking. It opens up exciting possibilities for future research and potential therapeutic advancements. While our current study is centered on the fundamental evolutionary biology of the IL-1 family, which requires a substantial amount of our time and computing resources to fully explore, we are already contemplating future research directions. We will be carefully considered when we plan our subsequent studies. We apologize for not being able to address this suggestion in our current study, but rest assured that it will be an important consideration for our future endeavors. Thank you for your understanding and for contributing such a valuable suggestion.
Reviewer 2 Report
Comments and Suggestions for Authors
The authors provided an through analysis of IL1s/IL1Rs across different species. The description of the topic is comprehensive. The evolutionary biology analysis is solid. However, I would like to suggest a significant improvement over the figure quality. The detailed suggestion is as follows:
Figure 1
The receptor subunits for IL1Rs are shown across the cell membrane. Does this mean there is one repeat for intracellular domain and three repeats for the extracellular domain for most of the IL1Rs? The length of the subunits are all equal. Does this mean each subunits share the same length? On the other hand, do receptors drawn close to each other form dimers? Please clarify or make modification accordingly.
Figure 2.
This figure contains too much information. For example, Figure 2B and Figure 2F are not mentioned in the text. It is also confusing for Figure 2C and Figure 2G. The cluster legend are invading into other figures. And some of the text is too small to read.
Figure 3.
It appears there is some non-readable text in the tree graph. Please either remove them or make them bigger. It is also not clear to me what the coloring in the inner circle of the tree graph stand for as well as the coloring of the outer circle. It will be nice if the author can direct label the Figure 3C and 3D in the graph instead of describing it.
Author Response
Comments 1: Figure 1. The receptor subunits for IL1Rs are shown across the cell membrane. Does this mean there is one repeat for intracellular domain and three repeats for the extracellular domain for most of the IL1Rs? The length of the subunits are all equal. Does this mean each subunits share the same length? On the other hand, do receptors drawn close to each other form dimers? Please clarify or make modification accordingly.
Response 2: We sincerely appreciate this observation and acknowledge that the original figure did not accurately represent the receptor-ligand system. Accordingly, we have revised the figure by removing the intracellular TIR domain from IL-1R2, which it does not possess. Additional clarifications regarding these modifications have been provided in the figure legend (see Figure 1) and detailed in the text (Lines 59-63).
Comments 2: Figure 2. This figure contains too much information. For example, Figure 2B and Figure 2F are not mentioned in the text. It is also confusing for Figure 2C and Figure 2G. The cluster legend are invading into other figures. And some of the text is too small to read.
Response 2: We have updated the manuscript by adding citations for the original Figure 2B and 2F (now renumbered as 2E and 2H) in Lines 231 and Lines 250-253. Additionally, we have reorganized Figure 2 according to the sequence of appearance, optimized font sizes for better readability, and particularly enhanced Figures 2A and 2C with revised captions to clearly present the main evolutionary groupings and their corresponding gene/protein sequence distributions. Please refer to Figure 2 and the accompanying explanations in Lines 274-289 for these improvements.
Comments 3: Figure 3. It appears there is some non-readable text in the tree graph. Please either remove them or make them bigger. It is also not clear to me what the coloring in the inner circle of the tree graph stand for as well as the coloring of the outer circle. It will be nice if the author can direct label the Figure 3C and 3D in the graph instead of describing it.
Response 3: We have revised Figure 3 by removing unclear Class labels and species names that were not referenced in the text. Additionally, we have annotated the color-coded tip points in the phylogenetic tree (Lines 400-403). Following your suggestions, we have labeled key conserved features directly in Figures 3C and 3D while removing redundant textual descriptions to improve clarity. Please refer to Figure 3 for full details.
Reviewer 3 Report
Comments and Suggestions for Authors
IL-1 superfamily across 400+ species: therapeutic targets and disease implications.
The authors have identified that both lineage-specific adaptations and evolutionarily conserved residues is essential for the design of targeted therapies, while also identifying less recognized species as valuable models for understanding early immune systems, thereby establishing a foundation for more effective therapeutics related to the IL-1 family in various diseases.
The introduction is well written, and the methods are clearly described.
Nonetheless, some major points that need addressing are listed below.
- Figure 2B is not referenced in the results section of the text.
- Same with Figure 2F. It is not described in the results’ text.
- In the Result section, all figures are typically introduced in order (chronologically) to avoid confusion. Currently, figures 2E and 2G (line 214) have been introduced before figure 2D (line 228). This makes it difficult to follow the text with the figures. Please reorder the figures such that the text introduces them one after the other. Once introduced sequentially, they can be used in any sequence for subsequent references.
- Figure 2H is not mentioned in the results section of the text. Missing from the text.
- The figure legends are confusing, as they are described in a non-sequential manner. For instance, Figs 2 A and E are described together (line 245), then B and F and so on. Please reorganize either the figures or the legends to ensure they correspond appropriately.
- Line 284 – which figure it is?
- Indicate Figure number for line 321.
- The entire first paragraph under “3. Main cross-species characteristics of IL-1s and IL-1Rs” (Lines 318 – 338) – require figure referencing. No figures have been cited for the same.
- The second paragraph (Lines 339-370) in the same result section also lacks figure references. Other than Table S1, no figures or tables have been mentioned.
- Line 376 – which figure?
- Figure number for line 379 is missing
- Section 3.4 lacks figure references, just like section 3.3. Is the entire paragraph (Lines 403-422) is about Fig S1? Please specify that accordingly.
- Same for lines 440-446
- Line 448 – “Our findings demonstrate that these domains are conserved in nearly all mammalian species [82].” – Why is this statement followed by a citation? Are the findings of this paper also explained in reference# 82? In other words, are the results of this section same as those of #82?
The Results section is excessively lengthy. Several of the paragraphs in the result section may be rewritten to solely highlight the findings, instead of discussing the field (which should be a part of the Discussion section). Further, relevant figures must always be cited in the Result section, as the absence of these references complicates the understanding of the results.
Similarly, the Discussion section is extensive. The findings of this paper are important, however the writing should maintain a primary literature focus, as it currently appears more like a review paper.
Author Response
Comments 1: Figure 2B is not referenced in the results section of the text. Same with Figure 2F. It is not described in the results’ text.
Response 1: We are profoundly grateful for your meticulous review of our manuscript. Your expert suggestions have been invaluable in significantly enhancing both the scholarly rigor and logical coherence of our work. The constructive criticism has helped us elevate the quality of our scientific reporting to meet the highest academic standards. We have updated the manuscript by adding citations for the original Figure 2B and 2F (now renumbered as 2E and 2H) in Lines 231 and Lines 250-253.
Comments 2: In the Result section, all figures are typically introduced in order (chronologically) to avoid confusion. Currently, figures 2E and 2G (line 214) have been introduced before figure 2D (line 228). This makes it difficult to follow the text with the figures. Please reorder the figures such that the text introduces them one after the other. Once introduced sequentially, they can be used in any sequence for subsequent references.
Response 2: We have reorganized Figure 2 by rearranging all panels in their order of appearance and sequentially adding corresponding figure legends, while simultaneously updating all relevant in-text citations throughout the manuscript. These revisions are reflected in Figure 2 and further detailed in Lines 274-289.
Comments 3: Figure 2H is not mentioned in the results section of the text. Missing from the text.
Response 3: We have added citations for the new panel (now labeled Figure 2I) at Lines 258, 314-315, 317, 472, and 476 to ensure proper referencing throughout the text.
Comments 4: The figure legends are confusing, as they are described in a non-sequential manner. For instance, Figs 2 A and E are described together (line 245), then B and F and so on. Please reorganize either the figures or the legends to ensure they correspond appropriately.
Response 4: We have reorganized the figure legends in Lines 274-289 to correspond with the new panel arrangement in Figure 2, ensuring each subfigure description matches its current position and numbering.
Comments 5: Line 284 – which figure it is?
Response 5: See Lines 315-317.
Comments 6: Indicate Figure number for line 321.
Response 6: See Line 425.
Comments 7: The entire first paragraph under “3. Main cross-species characteristics of IL-1s and IL-1Rs” (Lines 318 – 338) – require figure referencing. No figures have been cited for the same.
Response 7: We have introduced a new Figure 4 to clearly illustrate the key domain architectures and proportional distributions of both IL-1s and IL-1Rs. See Lines 422-443, Figure 4.
Comments 8: The second paragraph (Lines 339-370) in the same result section also lacks figure references. Other than Table S1, no figures or tables have been mentioned.
Response 8: See Lines 453-493
Comments 9: Line 376 – which figure?
Response 9: See Line 499.
Comments 10: Figure number for line 379 is missing
Response 10: See Lines 503-504.
Comments 11: Section 3.4 lacks figure references, just like section 3.3. Is the entire paragraph (Lines 403-422) is about Fig S1? Please specify that accordingly. Same for lines 440-446
Response 11: We have added the relevant figure citations in Lines 527-643 to properly reference this section.
Comments 12: Line 448 – “Our findings demonstrate that these domains are conserved in nearly all mammalian species [82].” – Why is this statement followed by a citation? Are the findings of this paper also explained in reference# 82? In other words, are the results of this section same as those of #82?
Response 12: We have added a comparative analysis between our findings and the results from the cited literature in Lines 622-625.
Comments 13: The Results section is excessively lengthy. Several of the paragraphs in the result section may be rewritten to solely highlight the findings, instead of discussing the field (which should be a part of the Discussion section). Further, relevant figures must always be cited in the Result section, as the absence of these references complicates the understanding of the results.
Response 13: We have thoroughly revised the latter portion of the Results section, systematically incorporating all relevant figure citations while removing any discussion-oriented content that was inappropriately placed in this section. These comprehensive updates can be found in Lines 526-643.
Comments 14: Similarly, the Discussion section is extensive. The findings of this paper are important, however the writing should maintain a primary literature focus, as it currently appears more like a review paper.
Response 14: We have substantially revised the Discussion section to focus on summarizing and contextualizing our key findings, while rigorously removing extraneous review-like content that diluted our core arguments. These refinements appear in Lines 647-649, 655-657, 664-749, 760-767, 773-776, 785-787, 889-891, and 935-938.
Round 2
Reviewer 2 Report
Comments and Suggestions for Authors
The graphs presented are of great quality. I have no further comments.
Reviewer 3 Report
Comments and Suggestions for Authors
The authors have successfully addressed my suggestions.